# River Extraction under Bankfull Discharge Conditions Based on Sentinel-2 Imagery and DEM Data

Dan Li [1,2], Ge Wang [1], Chao Qin [1] and Baosheng Wu [1,*]

1 State Key Laboratory of Hydroscience and Engineering, Tsinghua University, Beijing 100084, China; lidan985@mail.tsinghua.edu.cn (D.L.); wangg19@mails.tsinghua.edu.cn (G.W.); glqinchao@nwsuaf.edu.cn (C.Q.)
2 Emergency Science Research Institute, China Coal Research Institute, Beijing 100013, China
* Correspondence: baosheng@tsinghua.edu.cn

**Abstract:** River discharge and width, as essential hydraulic variables and hydrological data, play a vital role in influencing the water cycle, driving the resulting river topography and supporting ecological functioning. Insights into bankfull river discharge and bankfull width at fine spatial resolutions are essential. In this study, 10-m Sentinel-2 multispectral instrument (MSI) imagery and digital elevation model (DEM) data, as well as in situ discharge and sediment data, are fused to extract bankfull river widths on the upper Yellow River. Using in situ cross-section morphology data and flood frequency estimations to calculate the bankfull discharge of 22 hydrological stations, the one-to-one correspondence relationship between the bankfull discharge data and the image cover data was determined. The machine learning (ML) method is used to extract water bodies from the Sentinel-2 images in the Google Earth Engine (GEE). The mean overall accuracy was above 0.87, and the mean kappa value was above 0.75. The research results show that (1) for rivers with high suspended sediment concentrations, the water quality index (SRMIR-Red) constitutes a higher contribution; the infrared band performs better in areas with greater amounts of vegetation coverage; and for rivers in general, the water indices perform best. (2) The effective river width of the extracted connected rivers is 30 m, which is 3 times the image resolution. The $R^2$, root mean square error (RMSE), and mean bias error (MBE) of the estimated river width values are 0.991, 7.455 m, and $-0.232$ m, respectively. (3) The average river widths of the single-thread sections show linear increases along the main stream, and the $R^2$ value is 0.801. The river width has a power function relationship with bankfull discharge and the contributing area, i.e., the downstream hydraulic geometry, with $R^2$ values of 0.782 and 0.630, respectively. More importantly, the extracted river widths provide basic data to analyze the spatial distribution of bankfull widths along river networks and other applications in hydrology, fluvial geomorphology, and stream ecology.

**Keywords:** Sentinel-2 imagery; bankfull discharge; downstream hydraulic geometry; machine learning; Google Earth Engine; river width

## 1. Introduction

Currently, water resource availability is severely deficient, and the protection of water source areas and estimation of changes in runoff are receiving considerable attention. Rivers and streams are essential parts of the global hydrologic cycle; 90% of the water flux transported from continents to the ocean (approximately 37% of the total terrestrial precipitation) is carried by rivers [1,2]. River discharge and width are essential hydraulic variables and hydrological data needed to inform river management and restoration efforts. Bankfull discharge is morphologically critical because it represents the link between within-bank processes and floodplain processes, and bankfull discharge is frequently used to estimate the channel forming or dominant discharge of alluvial rivers [3,4]. Therefore, understanding the dynamics of bankfull discharge and bankfull river width at fine spatial

resolutions is essential for applications in hydrology, fluvial geomorphology, and stream ecology [5].

Remote sensing is an effective method for extracting open-surface inland water bodies over a variety of spatiotemporal scales compared with other field survey methods employed in the past decades [6–13]. Among all the extraction methods, the water index method is widely used to detect open-surface water bodies. In particular, the modified normalized difference water index (mNDWI) proposed by Xu [14] has been used by many researchers to extract global water bodies [15,16]; however, this technique produces errors in mixed pixels with water bodies and other land-cover types [17]. To reduce the effects of other land-cover types on the successful identification of water bodies, researchers have improved the accuracy of water detection by simultaneously utilizing several water body indices [18–20]. Wang et al. combined the mNDWI, land surface water index (LSWI), and two greenness-based vegetation indices (enhanced vegetation index (EVI) and normalized difference vegetation index (NDVI)) to detect open-surface water bodies [21]. Many researchers have conducted large-scale, rapid river extraction, and detected river channel changes based on the Google Earth Engine (GEE) platform [16,21–24]. GEE integrates many open source satellite images and various derivative products, providing strong support for efficient water body extraction [20]. However, for rivers with high suspended sediment concentrations, the existing methods are usually less accurate in river width extraction. The Yellow River has the highest suspended sediment concentration in the world. To extract water bodies from the Yellow River with high accuracy, new data and algorithms are needed to improve upon previously implemented methods.

River width datasets of various spatial resolutions, ranging from the global scale to the basin scale, have recently been provided by many researchers. Allen and Pavelsky provided the Landsat-derived North American River Width (NARWidth) dataset, which contains river width at mean annual discharge and extrapolates the strong relationship observed between the river width and its total surface area [25]. Allen and Pavelsky provided the first global river width database under mean annual discharge conditions based on Landsat imagery and found that rivers and streams likely play a great role in controlling land–atmosphere fluxes [26]. Li et al. extracted small and open-surface river information in the upper Yellow River by fusing a digital elevation model (DEM) and Sentinel-2 imagery corresponding to the average discharge during the summer flood season [20]. Gleason et al. extracted instantaneous cross-sectional flow widths during mean daily flow conditions from Landsat imagery and used these measurements to approximate the at-many-stations hydraulic geometry (AMHG) of the area; then, an ensemble of genetic algorithms was used to retrieve the instantaneous river discharge for each satellite image acquisition date [27]. Bankfull discharge is often used as a surrogate for channel forming or dominant discharge, which is the morphologically significant discharge that shapes the river [3,28]. Bankfull river width is one of the fundamental measures of stream size, and it is also a key parameter in the study of river geomorphology. However, most existing river width datasets were not acquired under bankfull discharge conditions due to (1) the limited number of high-quality images constrained by the satellite revisitation periods and the influence of snow and clouds; and (2) limited in situ measurement data can be used to calculate bankfull discharge. Therefore, attention should be given to river width extraction under bankfull discharge to better understand the river morphology and sediment transport spatial distributions, such as in the northeast Qinghai-Tibetan Plateau (QTP) region.

Yamazaki et al. developed the Global Width Database for Large Rivers (GWD-LR) for rivers wider than 183 m by applying a new algorithm to the SRTM Water Body Database and the HydroSHEDS flow direction map [29]. Pavelsky provided a river width dataset of Tanana for rivers wider than 150 m and calculated the power law relationship between the river widths and discharge amounts [30]. Allen and Pavelsky proposed the NARWidth dataset, which contains measurements of >2.4 $\times$ 10$^5$ km for rivers wider than 30 m. The researchers extrapolated the strong relationship observed between river width and the

total surface area measured at different river widths ($r^2 > 0.99$ for 100–2000 m widths) to narrower rivers and streams [25]. Pekel et al. mapped the global surface water and global hydromorphic features observed by Landsat satellites with a 30-m resolution over the past 32 years [15]. Allen and Pavelsky provided the first detailed global river width database for rivers wider than 90 m [26]. Based on the classification of river size proposed by Meybeck et al. [31], most of the existing river width datasets focus on medium to large rivers (small, medium, and large rivers have widths of 40–200, 200–800, and 800–1500 m, respectively). Very few studies have focused on small to medium river width estimations in mountainous areas using high-resolution satellite images [32–35]. As the headwaters of many large rivers, there is an abundance of small rivers with widths less than 200 m and smaller rivers with widths less than 40 m in the QTP. Therefore, a bankfull river width dataset with a finer resolution (river width < 90 m) for the QTP is necessary to facilitate research on fluvial geomorphology and hydrological modeling.

To date, there are no in-depth studies of the river network structure and runoff characteristics of the QTP, which is known as the "Asian Water Tower", containing the headwaters of ten major rivers across the Asian continent, and related research has just begun [19,34–36]. With the development of global climate change, the QTP has received more attention, and there is an urgent need for high-precision extraction of small rivers to understand the dynamic changes and hydraulic geometric relationships of rivers in this mountainous area. However, most river discharge and width observations are measured at ground-based gauges in the QTP region [36], which may limit a deeper understanding of river morphology, sediment transport, and flood routing, as well as their ecological impacts on the QTP.

In terms of the aforementioned research gaps, regarding the low width extraction precision of rivers with high suspended sediment concentrations, the less established physical meaning of the extracted river widths at annual mean discharge and mean discharge of summer flood season conditions, and the fact that the minimum extracted river widths are usually >90 m, the following objectives are proposed in this research: (1) obtain the river widths in the upper Yellow River Basin under bankfull discharge, combining a DEM, in situ hydrological data, and Sentinel-2 images with high temporal (5 days) and spatial resolutions (10 m); and (2) detect open-surface water bodies and monitor the downstream dynamic changes of river width under bankfull discharge conditions on the upper Yellow River Basin. A detailed technical flowchart is shown in Figure 1.

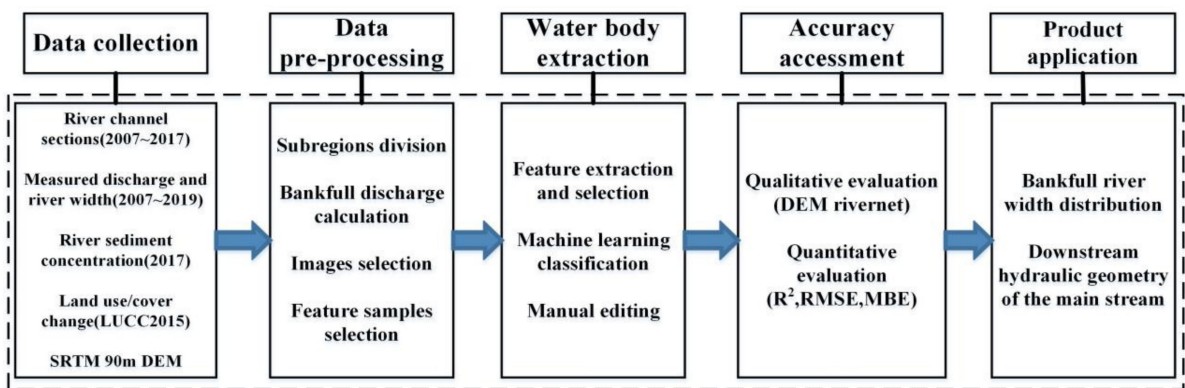

**Figure 1.** The technical flowchart of river extraction under bankfull discharge.

Considering the classification results of Meybect et al. [31] and the measured data from hydrological stations in the upper Yellow River Basin, rivers with widths smaller than 90 m are seen as small rivers in this study. It is anticipated that this research can close gaps in areas lacking hydrological data and assist in understanding the changes in river geometry within river networks.

## 2. Study Area and Data Preprocessing

### 2.1. Study Area

The study area is in the upper Yellow River (upstream from the Anningdu hydrological station) on the northeastern margin of the QTP. The total drainage area is approximately 250,944.65 km² (Figure 2). This area is composed of a series of alternating mountains, valleys, and hills with an elevation range of 1344–6295 m. The elevation shows a decreasing trend from more than 6200 m in the source area to less than 1400 m in the northeastern area. There are abundant river landforms and erosional types and many river canyons. A series of cascade reservoirs, such as Longyangxia, Laxiwa, Liujiaxia, and Lijiaxia, are built on the mainstream.

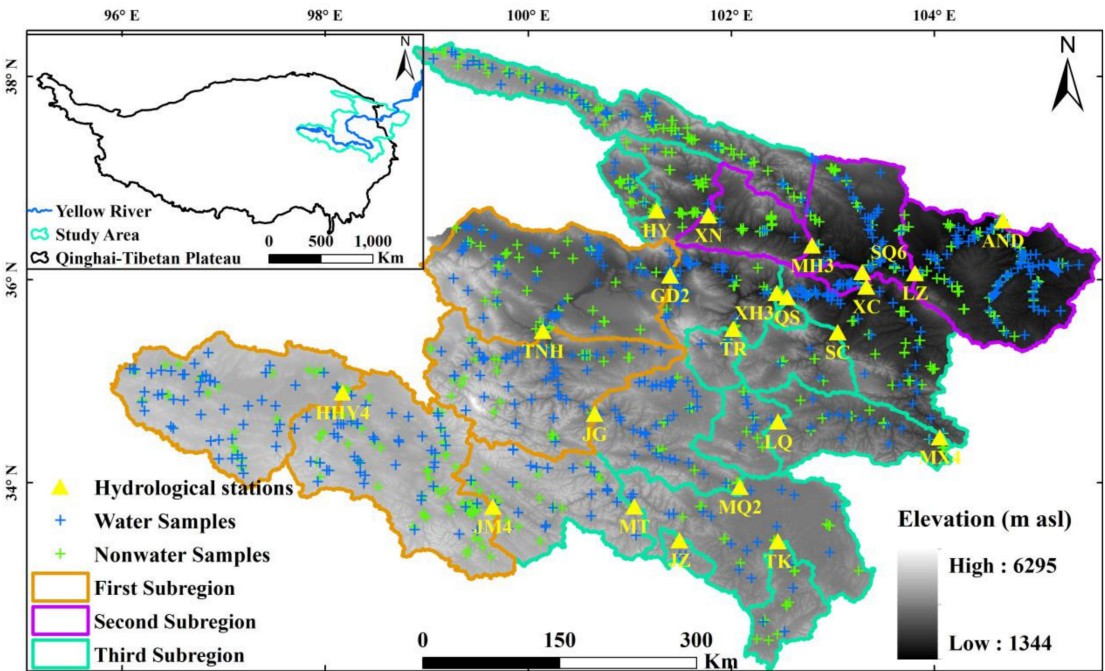

**Figure 2.** Geographical location of the study area and distribution of the hydrological stations and training samples. Subregions are marked as "First", "Second", and "Third". Hydrological stations are marked with yellow triangles. Water samples are marked with blue crosses, and nonwater samples are marked with green crosses.

The upper Yellow River is in the mid-latitude area, which has a typical plateau continental climate with little rain in winter and concentrated precipitation in summer. Controlled by the southwest and southeast air currents, the distribution of precipitation during the year is uneven, with large interannual variability. Approximately 60–80% of precipitation is concentrated from June to September, with the least precipitation occurring in December and January. Climate changes with terrain height lead to differences in precipitation and temperature throughout various regions, with small annual temperature differences, large daily temperature differences, long sunshine hours, and strong solar radiation days.

The suspended sediment concentrations of the different river sections in the study area are quite different; these conditions directly affect the extraction accuracy of river width. From June to October 2017, the suspended sediment concentration in the main stream of the Yellow River increased from 0.0456 kg/m³ at the HHY4 hydrological station at the source to 0.5552 kg/m³ at the TNH station; the concentration dropped rapidly to 0.0596 kg/m³ at the GD2 station, and then gradually increased to 1.3898 kg/m³ at the exit of the AND station. The reasons we use the suspended sediment concentration from June to October 2017 are (1) this time period corresponds to the Sentinel-2 image selection period, and the SSCs of the other time periods have no effects on the remote sensing images

selection; and (2) the sediment transport of the upper reaches of the Yellow River is mainly concentrated in June to October, accounting for 89–100% of the annual sediment transport.

The study area was divided into three subregions mainly on the basis of the surface features and river-suspended sediment concentrations. The cloud and snow cover conditions and mountain shadows were also considered (Figure 2). The underlying surface features and river-suspended sediment concentrations are the main criteria for subregional classification. The former is based on the national land-use cover change (LUCC2015) dataset collected in 2015, and the latter is based on the Annual Hydrological Reports of the People's Republic of China (2017) [37]. The main underlying surface types and river suspended sediment concentrations of the three subregions are shown in Table 1.

**Table 1.** Underlying surface types and water suspended sediment concentrations of the three subregions.

| Regions | Elevation (m) | NHS [1] | LUCC2015 [2] | SSC [3] (kg/m$^3$) | CSC [4] |
|---------|---------------|---------|--------------|--------------------|---------|
| First | 2177–6295 | 4 | 32, 33, 61, 62, 63, 65, 66, 67 | 0.046–0.556 | <20% |
| Second | 1344–4453 | 4 | 12, 32, 33, 51, 52, 65 | 0.199–1.390 | <5% |
| Third | 1629–5334 | 14 | 12, 22, 31, 32, 33, 52, 64, 67 | 0.055–0.818 | <5% |

[1] NHS—Number of hydrological stations; [2] LUCC—Land-use cover change; [3] SSC—Suspended sediment concentration; [4] CSC—Cloud and snow coverage; 12—Dryland, 22—Shrubland, 31/32/33—High/moderate/low coverage grassland, 51—Urban land, 52—Rural residential land, 61—Sandy land, 62—Gobi land, 63—Saline and alkaline land, 64—Wetland, 65—Bare land, 66—Bare rock land, and 67—Other unused land.

*2.2. Hydrological Data Collection and Processing*

2.2.1. Hydrological Data Collection

The data used in this study were acquired from the Annual Hydrological Reports of the People's Republic of China (1967–2019), and these data include river width, flow depth, flow velocity, flow discharge, suspended sediment concentration, and cross-section information. Referring to the research of Qin et al. [38], detailed information (river names, where the rivers flow to, longitude, latitude, altitude, contributing area, distance to estuary, cross sections selected to extract river widths, cross sections of the mainstream without reservoir effects, and annual peak discharge used to calculate flood frequency) of all 68 in situ-measured cross sections are presented in Table S1. The locations of the hydrological stations are represented with yellow triangles in Figure 2.

2.2.2. Bankfull Discharge Calculation

Bankfull river width is one of the basic channel geometry parameters associated with bankfull discharge. Therefore, the river width under the condition of bankfull discharge is the most significant in the river width extraction. In situ cross-sectional data (2007–2017) and in situ discharge data (1967–2019) from the upper Yellow River were used to determine the bankfull discharge of the main stream and its tributaries (Table S1).

Calculation of Bankfull Discharge Based on Cross-Section Morphology

In this study, cross sections that were less influenced by human activities (e.g., no hydropower stations or artificial diversions 10 km upstream or downstream of the measured cross section and located outside the backwater zone of a dam) and extreme events (e.g., glacial outbursts and landslides) were selected to maximize removal of external disturbances. The morphology of each cross section was determined based on in situ measurement data during 2007–2017.

The surveyed bankfull-stage indicator and its corresponding water level were detected for each cross section. Figure 3a shows a typical cross section of the main stream of the upper Yellow River at the Mentang (MT) hydrological station. The bankfull stage was obtained from the bankfull-stage indicator (red dot in Figure 3a) and the bankfull width was estimated from the surveyed cross-section geometry during 2007–2017. The bankfull discharge that corresponded to bankfull stage was obtained from either the measured data or stage–discharge relation (Figure 3b).

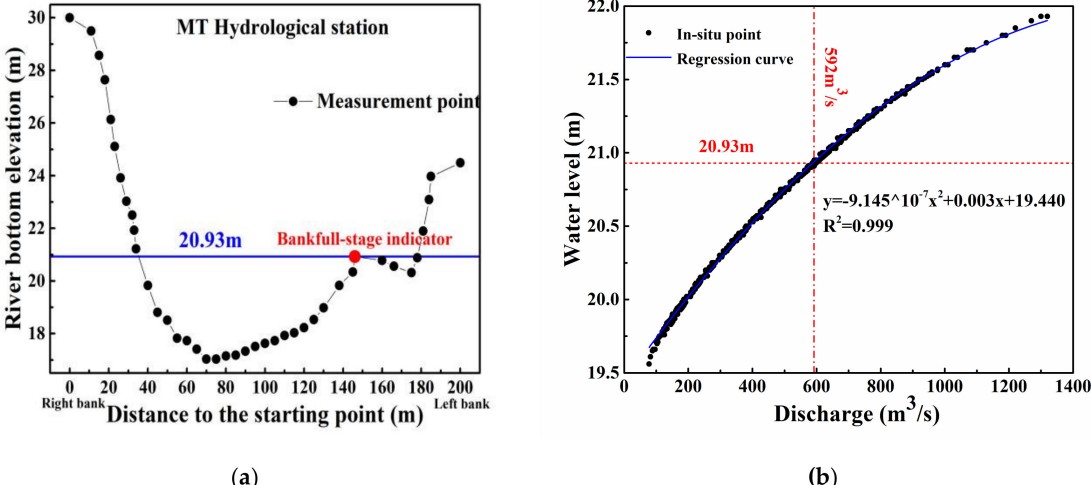

**Figure 3.** Cross-section morphology of the year 2015 (**a**) and the water level-discharge rating curve (**b**) of a typical cross section at the Mentang hydrological station.

Calculation of Bankfull Discharge Based on Flood Frequency

At many sections, the bankfull indicator is not available on mountainous river reaches, so bankfull discharges were determined based on flood frequency analysis. The annual maximum peak discharge of these cross sections was selected, and the Pearson III (P-III) curve was used to estimate the flood frequencies and corresponding discharges.

Two cases were used to determine the flood frequency and bankfull discharge. For the first case, there are more than two cross sections located within the same river reach. The morphology of each cross section was first depicted to see whether there is bankfull turning points (red point in Figure 3a). Then, the flood frequencies of all cross sections were estimated with P-III curves. For those cross sections that have no bankfull turning points, the flood frequencies of these cross sections were assumed to be the same as those cross sections with bankfull turning points. Lastly, the hydraulics (river widths and discharges) of the cross sections with no bankfull turning points under bankfull conditions were estimated through their shared flood frequency.

The second case occurs when there is only one cross section in the same river reach and when the cross section has no bankfull turning point. The flood frequency of the cross section was assumed to be the same as those cross sections with bankfull turning points within the same steam order. Then, the river width and discharge of the cross section were estimated under bankfull conditions through the shared flood frequency.

Referring to former studies of our research group [38,39] and comprehensively considering the remote sensing images coverage, we set up a criteria for hydrological data screening: (1) the completeness of the hydrological dataset (river width, discharge, and cross-section morphology); (2) have had relatively low anthropogenic influence (e.g., no hydropower station and artificial diversion 5 km upstream and downstream of the measured cross section, and located outside the backwater zone of a dam); (3) act as a natural riverway with perennial drainage; and (4) the quality and coverage of Sentinel-2 images under bankfull conditions. According to the above criterion, 22 hydrological stations (12 on the main stream and 10 on the tributaries) were ultimately selected from all 68 stations evaluated in this study (see Table S1). All 22 stations are located at single-thread river reaches though multi-thread reaches do exist in the upper Yellow River Basin. The bankfull discharges, calculation methods, and corresponding flood frequencies of the 22 hydrological stations are shown in Table 2.

**Table 2.** Bankfull discharges, river widths, and flood frequencies of 22 hydrological stations.

| Type | Station Name | Bankfull Discharge (m³/s) | Bankfull River Width (m) | Flood Frequency (%) | Method |
|---|---|---|---|---|---|
| **Mainstream** | Huangheyan4 (HHY4) | 54.5 | 87.5 | 48.8 | CSM [1] |
| | Jimai4 (JM4) | 419.0 | 149 | 87.0 | CSM |
| | Mentang (MT) | 592.0 | 143 | 87.0 | CSM |
| | Maqu2 (MQ2) | 1098.0 | 269.8 | 95.2 | Flood frequency |
| | Jungong (JG) | 1208.5 | 179 | 95.2 | Flood frequency |
| | Tangnaihai (TNH) | 1385.2 | 150.5 | 96.2 | Flood frequency |
| | Guide2 (GD2) | 1460.4 | 200 | 84.0 | Flood frequency |
| | Xunhua3 (XH3) | 1680.0 | 127.5 | 74.1 | CSM |
| | Xiaochuan (XC) | 1660.0 | 146 | 83.3 | CSM |
| | Shangquan6 (SQ6) | 1510.0 | 231.4 | 87.0 | Flood frequency |
| | Lanzhou (LZ) | 1880.0 | 206 | 87.0 | CSM |
| | Anningdu (AND) | 1720.0 | 162.5 | 87.0 | Flood frequency |
| **Tributary** | Qingshui (QS) | 22.0 | 29.1 | 49.5 | CSM |
| | Jiuzhi (JZ) | 54.7 | 46.6 | 89.6 | CSM |
| | Huangyuan (HY) | 48.9 | 34 | 38.8 | CSM |
| | Shuangcheng (SC) | 79.5 | 42.1 | 78.9 | CSM |
| | Luqu (LQ) | 155.7 | 38.9 | 20.6 | Flood frequency |
| | Tangke (TK) | 168.2 | 242 | 91.6 | Flood frequency |
| | Minxian4 (MX4) | 408.4 | 171.2 | 50.9 | Flood frequency |
| | Xining (XN) | 161.1 | 30.5 | 42.7 | Flood frequency |
| | Minhe3 (MH3) | 264.7 | 34.8 | 42.7 | Flood frequency |
| | Tongren (TR) | 98.3 | 34.4 | 69.9 | Flood frequency |

[1] CSM—Cross-section morphology.

### 2.3. Sentinel-2 Images Selection and DEM Processing

2.3.1. Sentinel-2 Images Selection under Bankfull Discharge

In order to be as close as possible to the time when the satellite passed through the hydrological stations, we chose the average discharge between 10 a.m. and 2 p.m. as the daily discharge (DD). To best represent the bankfull conditions, the dates on which the daily discharge falls within the bankfull discharge (BD × 1 ± 15%) interval were extracted from the in situ data of each of the 22 hydrological stations from June to October in 2017–2019. Then, Sentinel-2 multispectral instrument (MSI) images that can completely cover the contributing area of each hydrological station on the corresponding dates were selected. A website (https://scihub.copernicus.eu/userguide/, accessed on 21 February 2021) provides in-depth descriptions of the products and algorithms of Sentinel-2, as well as their performances. For individual hydrological stations with incomplete image cover or large amounts of cloud cover, the selection range of DD was expanded to BD × 1 ± 25% (Figure 4). Figure 4 shows the ratio relationship between the DD of the Sentinel-2 image cover and the BD of the 22 hydrological stations.

Figure 4 shows that 81.25% of the images have DD values within BD × 1 ± 15%. The images located in the range of BD × 1 ± 15% to BD × 1 ± 25% are mostly used to supplement the corner position of the control basins. The left side of the red line in the figure consists of images of the mainstream of the upper Yellow River, while the right side comprises images of the tributaries. The DD along the mainstream is closer to the BD than the tributaries.

2.3.2. River Network Extraction from DEM

Drainage network extraction plays an important role in geomorphologic analyses and hydrologic modeling studies, among other applications. Bai [40] and Wu et al. [41] used an enhanced flow enforcement method without elevation modification towards accurate and efficient drainage network extraction. In this study, the Drainage Network Extraction Tool (DNET) developed by Wu et al. [41] and Bai [40] was used to extract a total of seven orders

of river networks from the SRTM 90-m DEM according to the minimum confluence area of 7.29 km$^2$.

In addition to the rivers, there are other ground features in the study area, such as roads, buildings, vegetation, and bare land, which directly affect the efficiency and accuracy of river extraction. River networks with orders of 3–7 were used to construct buffer zones for the river networks according to a fixed distance, and the Sentinel-2 images under bankfull discharge conditions were masked using buffer zones. Through this operation, all the information outside the buffer zone of the river networks can be removed to reduce the impact of other noise. In this study, based on the measured river widths from the hydrological data and Google Earth, the buffer distance of the river networks with orders of 3–4 was set to 1000 m, and the buffer distance of the river networks with orders of 5–7 was set to 3000 m.

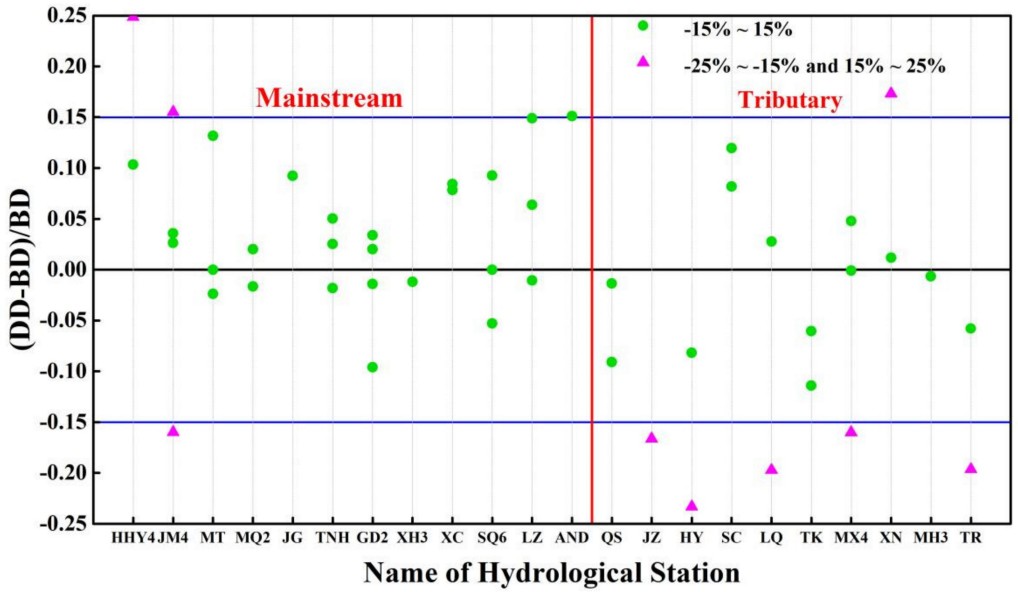

**Figure 4.** The ratios of the daily discharges to bankfull discharges.

## 3. Method of River Extraction

The underlying surface and suspended sediment concentrations of water bodies directly affect the river extraction accuracy and the data postprocessing workload [20]. Some problems exist when using the water body index threshold method to extract water bodies. When the threshold is too large, despite achieving basically complete water body extraction, too many noise points exist, and the data postprocessing workload is considerable; when the threshold is too small, there are fewer noise points, and the postprocessing workload is less intense, but the water body cannot be completely extracted. To improve the accuracy of the river extraction while also reducing the workload of data postprocessing, a machine learning (ML) random forest (RF) algorithm was used to extract the water bodies according to different underlying surface features and river-suspended sediment concentrations (Table 1). ML involves the use of data or past experiences to optimize the performance standards of computer programs. RF is a classification tree-based algorithm proposed by Breiman [42]. RF is essentially an extension of the traditional decision tree algorithm, and it improves the classification accuracy of models by combining multiple decision trees.

### 3.1. Training Samples Selection

Based on Sentinel-2 images under bankfull discharge in the control basin of 22 hydrological stations determined in Section 2.3, water and nonwater samples (including vegetation, residential land, roads, farmland, bare land, and snow) were manually selected from the land-cover images on the GEE platform. The first subregion has a total of 713 samples (400 water bodies, 313 nonwater bodies); the second subregion has a total

of 409 samples (202 water bodies, 207 nonwater bodies); and the third subregion has 608 samples (302 water bodies, 306 nonwater bodies) (Figure 2).

### 3.2. Features Extraction

Considering the characteristics of high altitude, a complex underlying surface, and a high suspended sediment concentration in the study area, three types of features were extracted from the DEM and Sentinel-2 imagery. These features are listed as follows:

- Basic information of the DEM and the band reflectivity of Sentinel-2 images (10 features) are provided, including elevation, aspect, slope, and hillshade derived from the DEM and the band reflectivities of B2, B3, B4, B8, B11, and B12 from the Sentinel-1 imagery.
- The gray level cooccurrence matrix (GLCM) is employed to derive certain textural features (180 features). GEE provides a total of 18 matrices, of which 14 are from Haralick et al. [43] and 4 are from Conners et al. [44]. Please refer to these two papers for the meaning of each feature and their detailed calculation formulas, as this study will not explain these features in detail. For the 10 basic features obtained in the first step, the 18 texture features were extracted from the GEE platform.
- The spectral indices of the Sentinel-2 images (49 features) were constructed based on the apparent reflectance of the B2, B3, B4, B8, B11, and B12 bands (see Table 3). Most of the spectral indices originate from the remote sensing index database (https://www.indexdatabase.de/, accessed on 21 February 2021).

**Table 3.** Spectral indices of the Sentinel-2 images.

| Spectral Indices | Formula | Reference |
|---|---|---|
| Normalized Difference Water Index | $NDWI = (B3 - B8)/(B3 + B8)$ | [45] |
| Modified Normalized Difference Water Index | $MNDWI = (B3 - B11)/(B3 + B11)$ | [46] |
| Normalized Difference Water Index 3 | $NDWI3 = (B8 - B11)/(B8 + B11)$ | [47] |
| Automated Water Extraction Index | ① $AWEIsh = B2 + 2.5 \times B3 - 1.5 \times (B8 + B12) - 0.25 \times B11$ <br> ② $AWEInsh = 4 \times (B3 - B12) - (0.25 \times B8 + 2.75 \times B11)$ | [9] |
| Enhanced Water Index | $EWI = (B3 - B8 - B12)/(B3 + B8 + B12)$ | [48] |
| Water Index 2015 | $WI2015 = 1.7204 + 171 \times B3 + 3 \times B4 - 70 \times B8 - 45 \times B11 - 71 \times B12$ | [49] |
| Revised Normalized Difference Water Index | $RNDWI = (B12 - B4)/(B12 + B4)$ | [50] |
| Shadow Water Index | $SWI = B2 + B3 - B8$ | [51] |
| Enhanced Shadow Water Index | $ESWI = (B2 + B3)/(B8 + B8)$ | [52] |
| New Comprehensive Water Index | $NCWI = (7 \times B3 - 2 \times B2 - 5 \times B8)/(7 \times B3 + 2 \times B2 + 5 \times B8)$ | [53] |
| New Water Index | $NWI = ((B2 - (B8 + B11 + B12))/(B2 + (B8 + B11 + B12))) \times 100$ | [54] |
| Normalized Difference Building-up Index | $NDBI = (B12 - B8)/(B12 + B8)$ | [55] |
| Normalized Difference Vegetation Index | $NDVI = (B8 - B4)/(B8 + B4)$ | [56] |

**Table 3.** *Cont.*

| Spectral Indices | Formula | Reference |
|---|---|---|
| Green Normalized Difference Vegetation Index | $GNDVI = (B8 - B3)/(B8 + B3)$ | |
| Ratio Vegetation Index | $RVI = B8/B4$ | |
| Enhanced Vegetation Index | $EVI = 2.5 \times (B8-B4)/((B8 + 6.0 \times B4-7.5 \times B2) + 1.0)$ | |
| Difference Vegetation Index | $DVI = B8 - B4$ | |
| Green Difference Vegetation Index | $GDVI = B8 - B3$ | |
| Weighted Difference Vegetation Index | $WDVI = B8 - 0.460 \times B4$ | |
| Renormalized Difference Vegetation Index | $RDVI = (B8 - B4)/(B8 + B4) \times 0.5$ | |
| Pan Normalized Difference Vegetation Index | $PNDVI = (B8 - (B3 + B4 + B2))/(B8 + (B3 + B4 + B2))$ | |
| Red-Blue Normalized Difference Vegetation Index | $RBNDVI = (B8 - (B4 + B2))/(B8 + (B4 + B2))$ | |
| Blue-Normalized Difference Vegetation Index | $BNDVI = (B8 - B2)/(B8 + B2)$ | |
| Blue-Wide Dynamic Range Vegetation Index | $BWDRVI = (0.1 \times B8 - B2)/(0.1 \times B8 + B2)$ | https://www. indexdatabase.de/, accessed on 21 February 2021 |
| Simple Ratio Red/NIR Ratio Vegetation-Index | $SRRed\_NIR = B4/B8$ | |
| Simple Ratio MIR/Red Eisenhydroxid-Index | $SRMIR\_Red = B12/B4$ | |
| Soil-Adjusted Vegetation Index mir | $SAVImir = (B8 - B12) \times (1.0 + 0.401)/(B8 + B12 + 0.401)$ | |
| Adjusted Transformed Soil-Adjusted Vegetation Index | $ATSAVI = 1.22 \times (B8 - 1.22 \times B4 - 0.03)/(1.22 \times B8 + B4 - 1.22 \times 0.03 + 0.08 \times (1.0 + 1.22 \times 2.0))$ | |
| Transformed Soil Adjusted Vegetation Index | $TSAVI = (0.743 \times (B8 - 0.743 \times B4 - 0.323))/(B4 + 0.743 \times (B8 - 0.323) + 0.413 \times (1.0 + 0.743 \times 2.0))$ | |
| | $PRWI = (B3 + B8)/(B3 - B8)$ | |
| Soil Composition Index | $SCI = (B11 - B8)/(B11 + B8)$ | |
| Ratio Drought Index | $RDI = B12/B8$ | |
| Moisture Stress Index 2 | $MSI2 = B11/B8$ | |
| Tasselled Cap-wetness | $WET = 0.1509 \times B2 + 0.1973 \times B3 + 0.3279 \times B4 + 0.3406 \times B8-0.7112 \times B11-0.4572 \times B12$ | |
| Normalized Burn Ratio | $NBR = (B8 - B12)/(B8 + B12)$ | |
| Simple Ratio 520/670 | $SR520\_670 = B2/B4$ | |
| Simple Ratio 550/800 | $SR550\_800 = B3/B8$ | |
| Simple Ratio 800/2170 | $SR800\_2170 = B8/B12$ | |
| Simple Ratio 800/550 | $SR800\_550 = B8/B3$ | |
| Simple Ratio 833/1649 MSIhyper | $SR833\_1649 = B8/B11$ | |
| Difference 678/500 | $D678\_500 = B4 - B2$ | |
| Visible Atmospherically Resistant Index Green | $VARIgreen = (B3 - B4)/(B3 + B4 - B2)$ | |
| Iron Oxide | $IO = B4/B2$ | |
| Ferric iron, $Fe^{2+}$ | $Fe2 = B12/B8 + B3/B4$ | |
| Ferric iron, $Fe^{3+}$ | $Fe3 = B4/B3$ | |
| Shape Index | $IF = (2.0 \times B4 - B3 - B2)/(B3 - B2)$ | |
| Coloration Index | $CI = (B4 - B2)/B4$ | |
| Redness Index | $RI = (B4 - B3)/(B4 + B3)$ | |
| Color Rendering Index 550 | $CRI550 = B2 \times (-1.0) - B3 \times (-1.0)$ | |
| Alteration | $Alteration = B11/B12$ | |

B2—Blue; B3—Green; B4—Red; B8—NIR; B11—MIR; B12—SWIR.

### 3.3. ML-RF Features Selection and Water Extraction

In this study, we first tested for multicollinearity between variables and removed variables with a Pearson's r > 0.85 to reduce the redundancy. However, the noise in the water body extraction results was much more than that of the extraction results without removing any variables, so we did not remove any variables in the end. The feature variables that substantially contributed to the model's accuracy were sorted according to the mean decrease accuracy (MDA) method in the RF algorithm (http://blog.datadive. net/selecting-good-features-part-iii-random-forests/, accessed on 21 February 2021). The MDA method directly measures the impact of each feature on the accuracy of the model. The main idea of this method is to disrupt the order of the features and measure the impact of order changes on the accuracy of the model. The order of the feature variables sorted from top to bottom is shown in Figure 5. The first 6 features were selected to represent all 239 features to construct a feature subset for ML modeling.

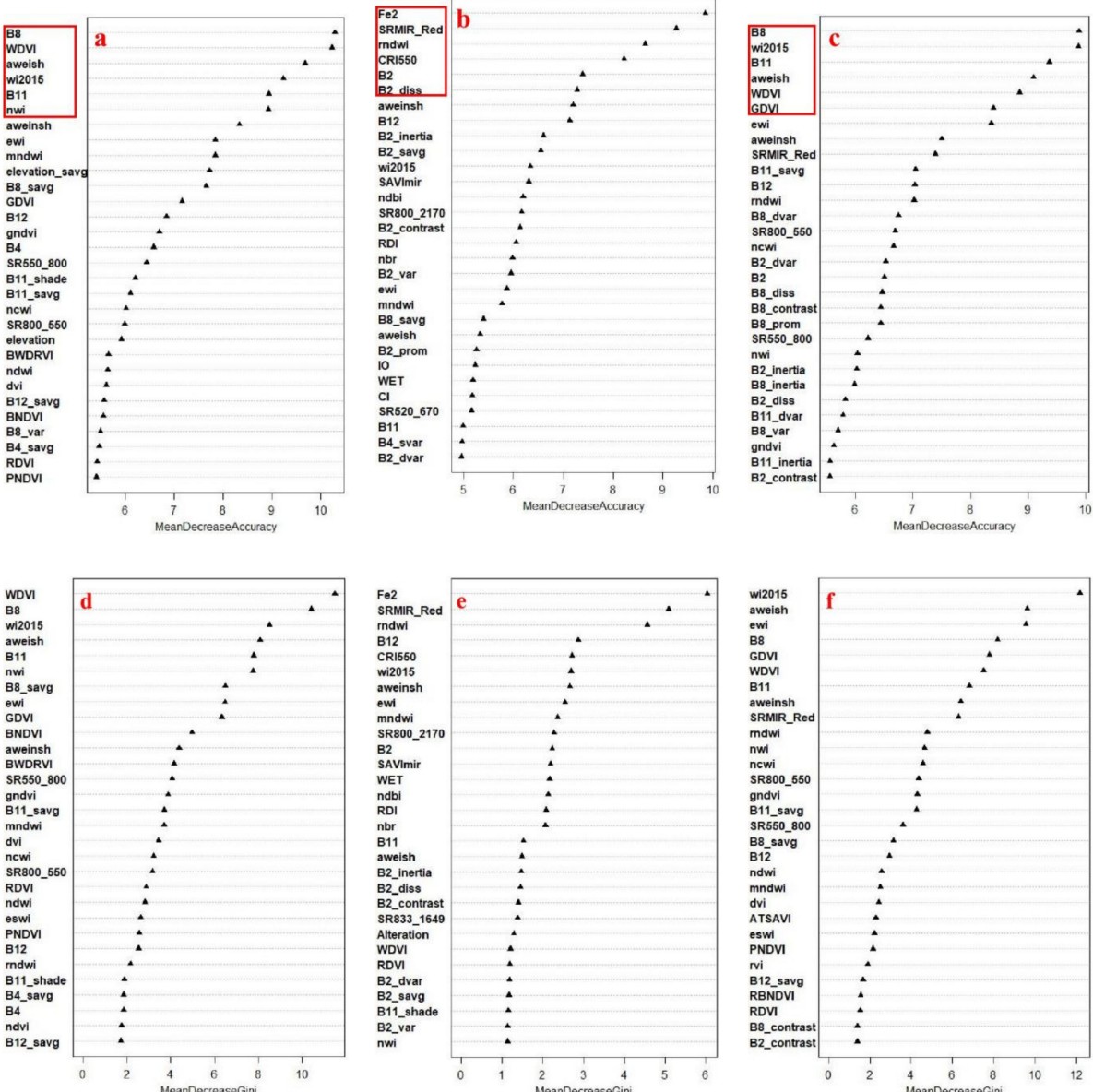

**Figure 5.** Ranking of the importance of the feature variables in three subregions. (**a–c**) ranking of MDA for the first, second, and third region; (**d–f**) ranking of MDG for the first, second, and third region.

The RF model was used to reorder the feature subsets of the three subregions according to the value of MDA and the impact of each feature on the Gini node impurity. The six features of the three subregions in order of importance are aweish, ewi, wi2015, B8, WDVI, and B11 in the first region; SRMIR_Red, B2, rndwi, B2_diss, CRI550, and Fe2 in the second region; and B8, B11, wi2015, WDVI, aweish, and GDVI in the third region.

The model-feature ranking shows that for the first region where the river-suspended sediment concentration is lowest, the underlying surface is mainly bare land (rocks, sandy land) with low-to-medium grassland coverage, and the top three contributing characteristics are the water body indices. For the second region, where the river-suspended sediment concentration is highest and the underlying surface features are mainly urban and rural land, arable land, and low-to-medium grassland coverage, the largest contribution is the water quality index-SRMIR_Red, followed by the blue band and water body index. This finding shows that the water quality index greatly contributes to this area with a high suspended sediment concentration and more bare yellow land. For the third region where the river suspended sediment concentration ranks in the middle and the underlying surface is mainly high-coverage grasses and shrubs, the two infrared bands (B8 and B11) contribute the most, which further supports the sensitive response of the infrared band to water and vegetation.

### 3.4. Model Evaluation

The overall accuracy (OA) and kappa coefficient were calculated from the confusion matrix to characterize the accuracy of the ML modeling classification. The kappa coefficient is a ratio that represents the error reduction between an evaluated classification and a completely random classification. In general, the minimum allowable discrimination accuracy of the kappa coefficient is 0.7 [57]. The formula is shown below:

$$K = \frac{N \cdot \sum_i^r x_{ii} - \sum(x_{i+} \cdot x_{+i})}{N^2 - \sum(x_{i+} \cdot x_{+i})} \tag{1}$$

where $K$ is the kappa coefficient, $r$ is the number of rows in the error matrix, $x_{ii}$ is the value on row $i$ and column $i$ (main diagonal), $x_{i+}$ and $x_{+i}$ are the sum of the *i-th* row and the *i-th* column, respectively, and $N$ is the total number of samples. In most cases, kappa statistics are used to evaluate the classification effect. Landis and Koch [58] determined that when the statistical kappa value is within the range of 0.60–0.80, the strength of agreement is substantial, and when the statistical kappa value is above 0.8, the strength of agreement is nearly perfect.

A total of 10 repeated models were executed, and the repeated 3-fold cross-validation results are shown in Table 4.

**Table 4.** The results of the 3-fold cross-validation.

| Region ID | OA Min | OA Max | OA Mean | OA SD | Kappa Min | Kappa Max | Kappa Mean | Kappa SD |
|---|---|---|---|---|---|---|---|---|
| First | 0.8654 | 0.8942 | 0.8788 | 0.0101 | 0.7292 | 0.7823 | 0.7530 | 0.0192 |
| Second | 0.8559 | 0.928 | 0.8993 | 0.0245 | 0.7095 | 0.8554 | 0.7978 | 0.0496 |
| Third | 0.8895 | 0.9379 | 0.8998 | 0.0189 | 0.7791 | 0.8749 | 0.8049 | 0.0292 |

Combining the information in Table 4 with Landis and Koch's [58] classification, the classification results of the three subregions are very good, and the classification OA is in the order of third > second > first; the kappa average value of the third subregion is >0.8.

### 4. Results

### 4.1. River Extraction Results

Although the drainage network from the DEM is used to constrain the image and remove most of the background noise, there is still strong background noise present in

certain areas. The extracted river raster results are converted into vectors using ARCGIS 10.2, and then, the individual areas with noise are edited manually. The final editing results are shown in Figures 6 and 7. Figure 6 shows the result of the extracted connected rivers above 30 m and the drainage network above order 3 from the DEM; Figure 7 shows the result of all the extracted rivers (connected + disconnected) and the drainage network above order 2.

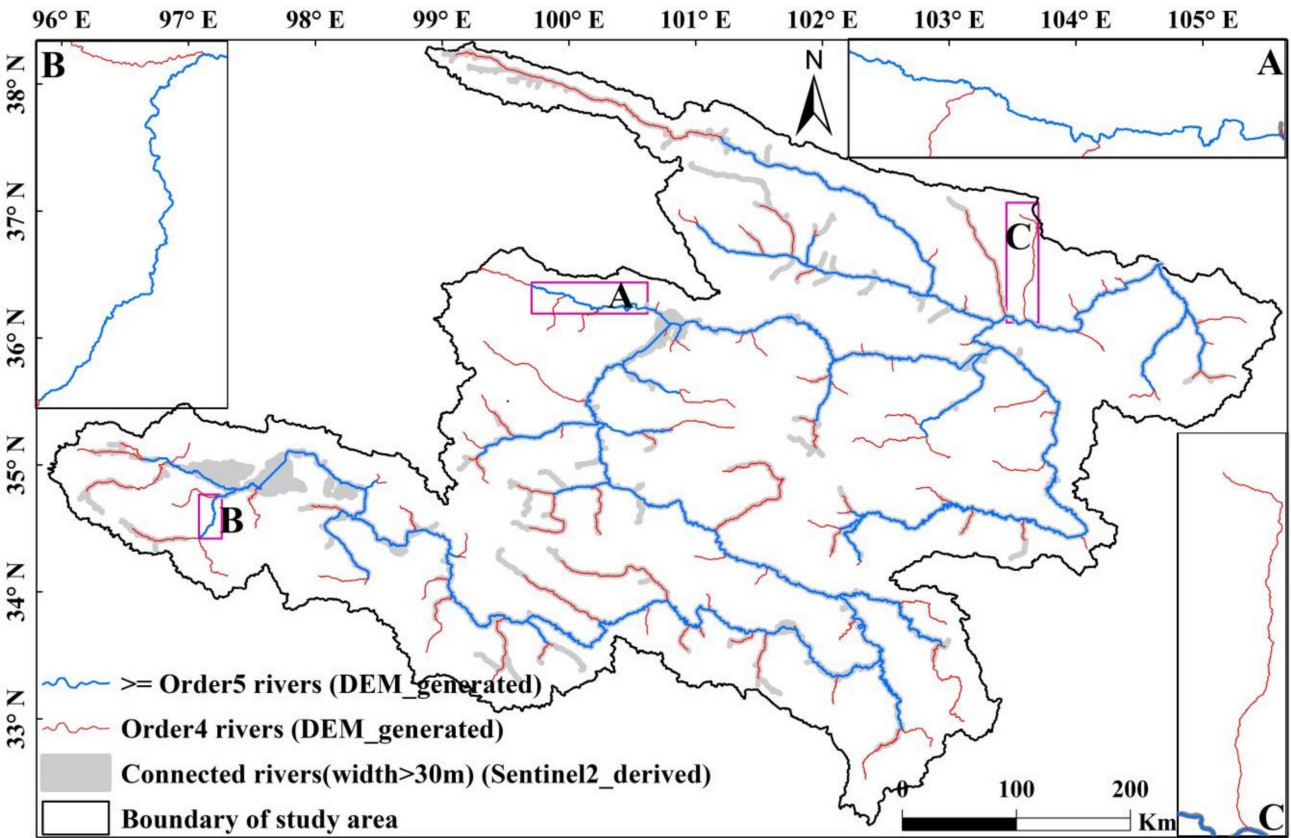

**Figure 6.** Comparisons between the extracted connected rivers with a width over 30 m and river networks above order 3 that were generated from a 90-m resolution DEM.

Figure 6 shows that all rivers (excluding areas A and B) above order 4 and nearly half of the order 4 rivers were extracted. The effective connected width of the extracted rivers is greater than 30 m, which is 3 times the image resolution. Figure 7 also shows that the rivers extracted by the ML method basically cover all the river networks above order 3 and most of the order 3 rivers in the 10-m Sentinel-2 images under bankfull discharge. Among them, area C was not successfully extracted; thus, the original satellite image was checked, and the area contained a township road. In and around area A, in the northern part of area B, and on the right side of area C, some rivers of orders 3 and 4 were not successfully extracted. Based on a check of the satellite images, one reason that this extraction was unsuccessful was that some river sections were too narrow (river width < 20 m); the other reason was due to image losses caused by cloud removal processing; and the third explanation was that the extraction result was too noisy to distinguish the boundary of the river and directly delete it.

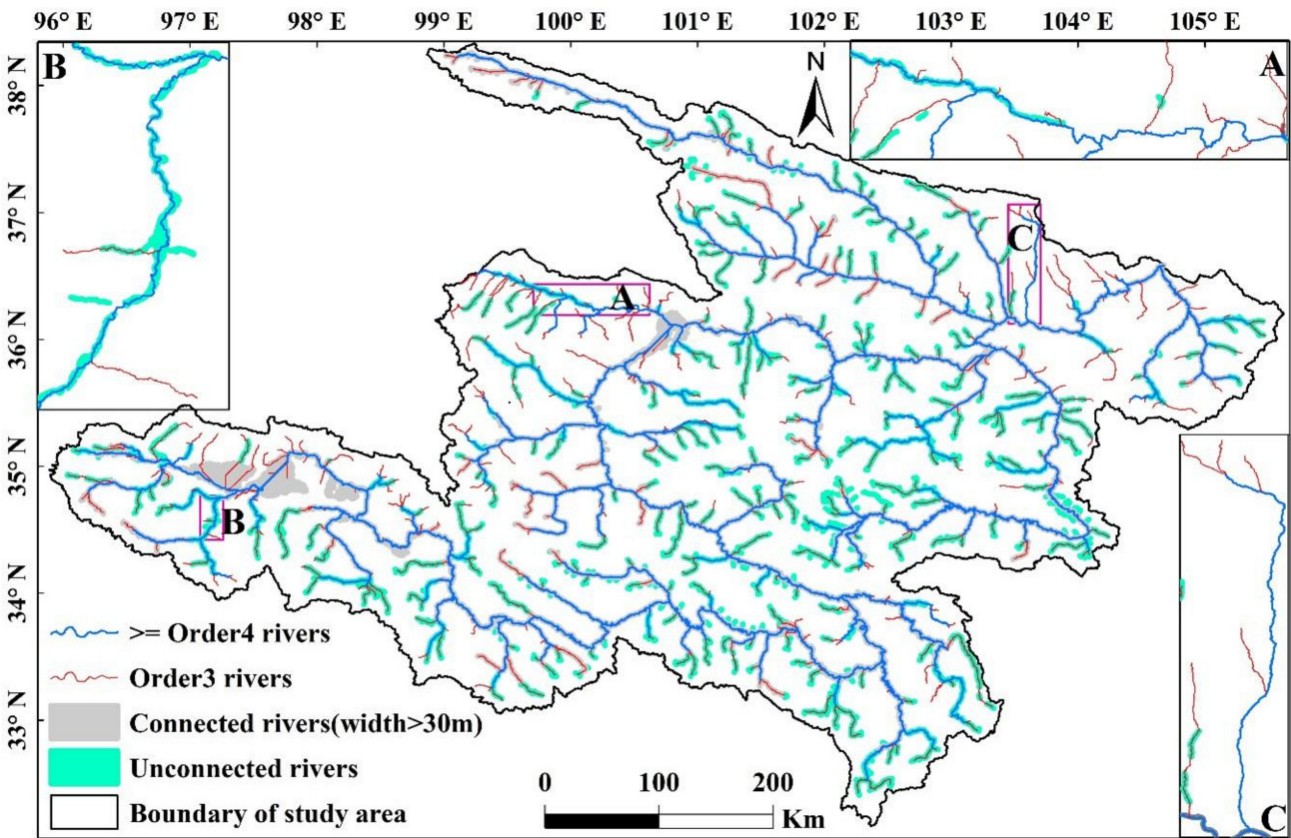

**Figure 7.** Comparisons between all the extracted rivers and drainage networks above order 2 that were generated from a 90-m resolution DEM.

### 4.2. Estimation Accuracy of River Width

Three indicators, i.e., $R^2$, root mean square error (RMSE), and mean bias error (MBE) [20], were used to quantitatively evaluate the estimated river width. The boundary of the extracted river vectors in Figure 6 was considered to be the bank of the river. First, the river center lines were extracted from the final edited river results, and then, the perpendicular lines of the river center lines were established [26]. There are two intersections between the perpendicular line and the two banks of the river. For a single-thread river, the distance between the intersections of the two banks of the river is the river width. For a multiple-thread river, the river width is the sum of the width of each flow. For each measured cross section, the resolution of the remote sensing image is 10 m; therefore, 5 m is taken as the step length, and the average of three consecutive measurements is taken as the final river width. Figure 8 shows the linear regression between the estimated bankfull river width and the measured river width from 22 hydrological stations.

The in situ river widths of 22 hydrological stations were used to evaluate the estimated river widths extracted from remote sensing images (Figure 8). The estimated river width results are satisfactory. The $R^2$, RMSE, and MBE results are 0.991, 7.455 m, and −0.232 m, respectively (Figure 8). The RMSE is calculated within one pixel, and the overall river width is underestimated. The results indicate that the estimated river widths of the single channel of the mainstream reaches are basically within 300 m.

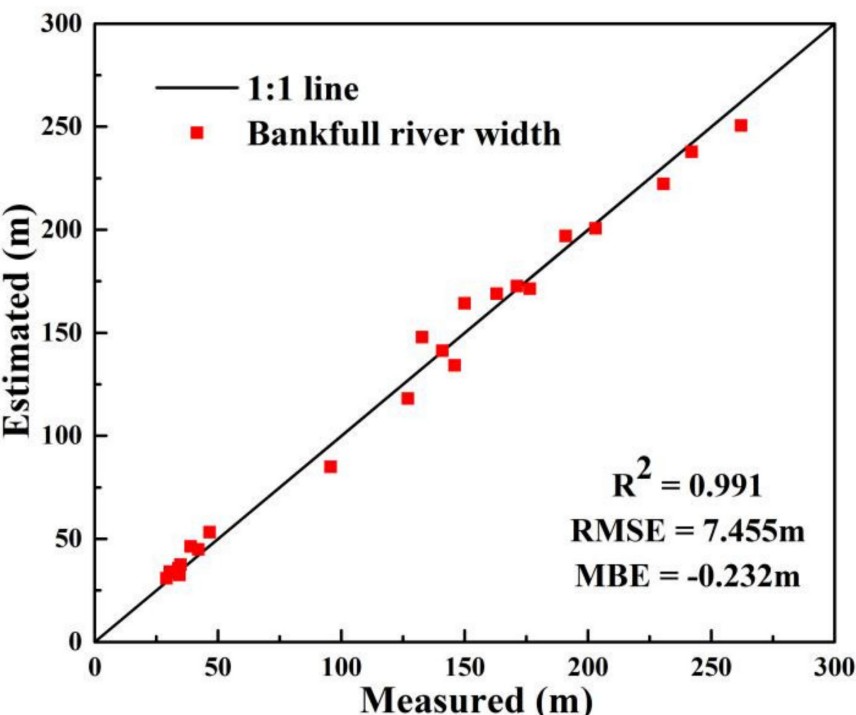

**Figure 8.** Comparison between the estimated river width and in situ-measured data.

### 4.3. River Width Distribution along the Mainstream of the Upper Yellow River

The machine learning method was used to extract the rivers in the upper Yellow River from the Sentinel-2 images through the GEE platform. Excluding some small headwaters at the source of the Yellow River, the main stream of the upper Yellow River is completely extracted. Within a distance of 2000+ km from the source of the extracted main stream to the exit of Anningdu station (AND), 5 km was taken as the step length to extract the river widths along the main stream, excluding lakes and reservoirs. The results are shown in Figure 9.

Figure 9 shows the distribution characteristics of the estimated bankfull river width along the main stream of the Yellow River (Figure 9a), as well as the increase in the contributing area and discharge of each hydrological station (Figure 9b). The single-thread river reaches and multithread river reaches are staggered. There are four single-thread river reaches (gray area, Figure 9a) and three multithread river reaches (light blue area, Figure 9a), though all 12 hydrological stations are in single-thread reaches.

Based on the estimated bankfull river width along the main stream, the mean and standard deviation (SD) of the river widths of the two river types were calculated with single threads and multiple threads in different river reaches. The average river width of the single-thread river reaches shows a stronger linear relationship along the main stream, with an $R^2$ of 0.801 (Figure 9a). The contributing area and discharge also gradually increase along the mainstream, and there are positive correlations between the bankfull river width and the contributing area and discharge. The average river widths of the multithread river reaches vary greatly and have no obvious regularity.

The estimated river widths, contributing areas, and bankfull discharges of the hydrological stations were selected to study their relationships in two situations: (1) single-thread river reaches, excluding the hydrological stations affected by reservoirs (Figure 10a,b); and (2) all river reaches, excluding the hydrological stations affected by reservoirs (Figure 10c,d).

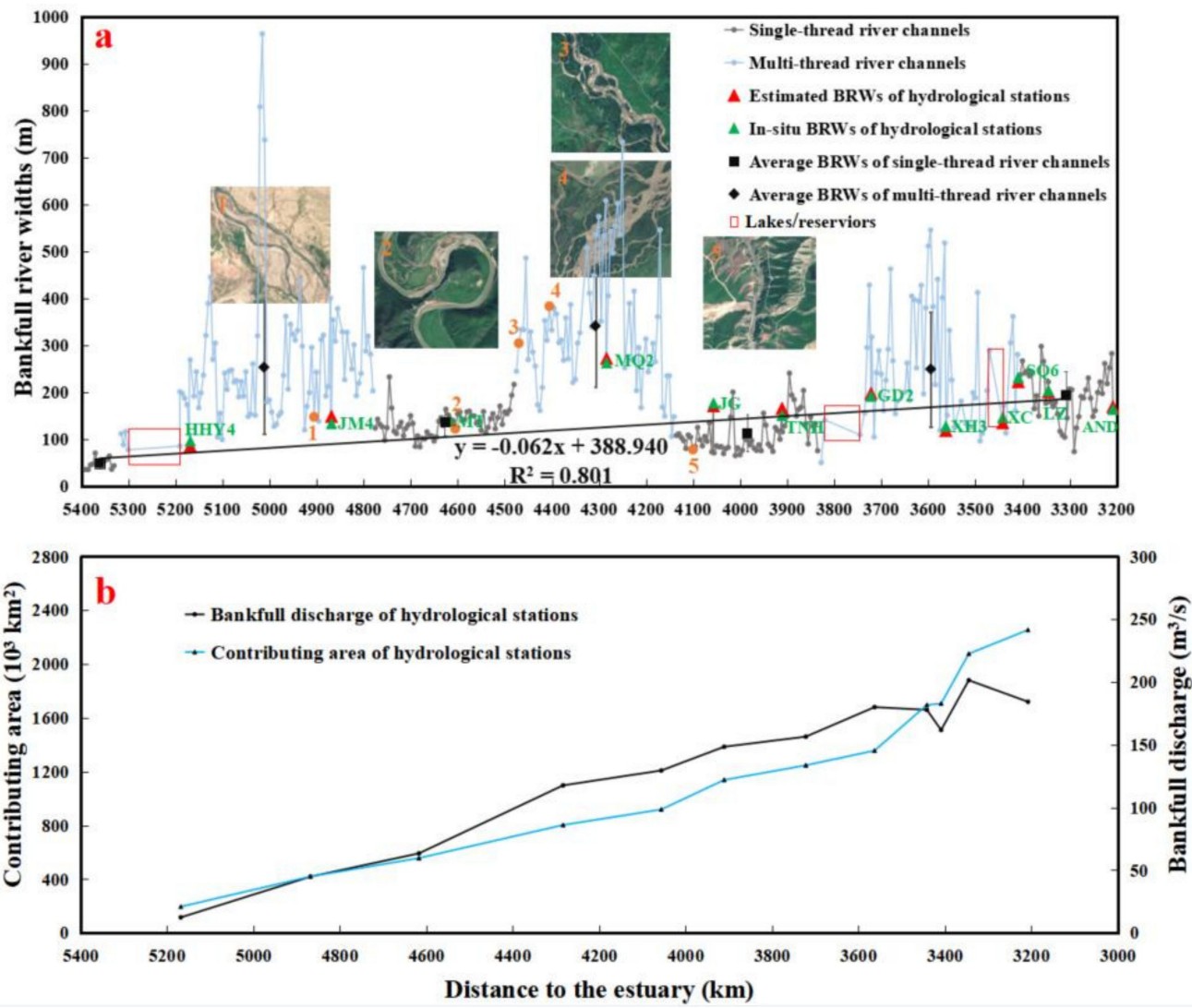

**Figure 9.** Distribution characteristics of the (**a**) bankfull river widths and (**b**) contributing area and bankfull discharge along the main stream of the upper Yellow River. BRW—Bankfull river width.

Figure 10b shows a regression curve between the estimated river widths and the bankfull discharges of the single-thread river reaches, representing a good downstream hydraulic geometry relationship, with an $R^2$ of 0.782. Since there is a positive correlation between the contributing areas and the bankfull discharges, a regression curve between the contributing area and river width is created, which is in the form of a power law, as shown in Figure 10a, and the $R^2$ is 0.630. In addition, the relationships between the river widths and the contributing areas and bankfull discharges of all hydrological stations less affected by reservoirs and human activities were determined, as shown in Figure 10c,d, with $R^2$ values of 0.462 and 0.662, respectively, which are slightly lower than those of the single-thread river reaches. The $R^2$ of river width versus bankfull discharge is greater than the $R^2$ of the river width versus contributing area relationship. This finding confirms the results of Wilkerson et al. [59], who stated that using the contributing area alone does not yield a reliable river width versus contributing area relationship.

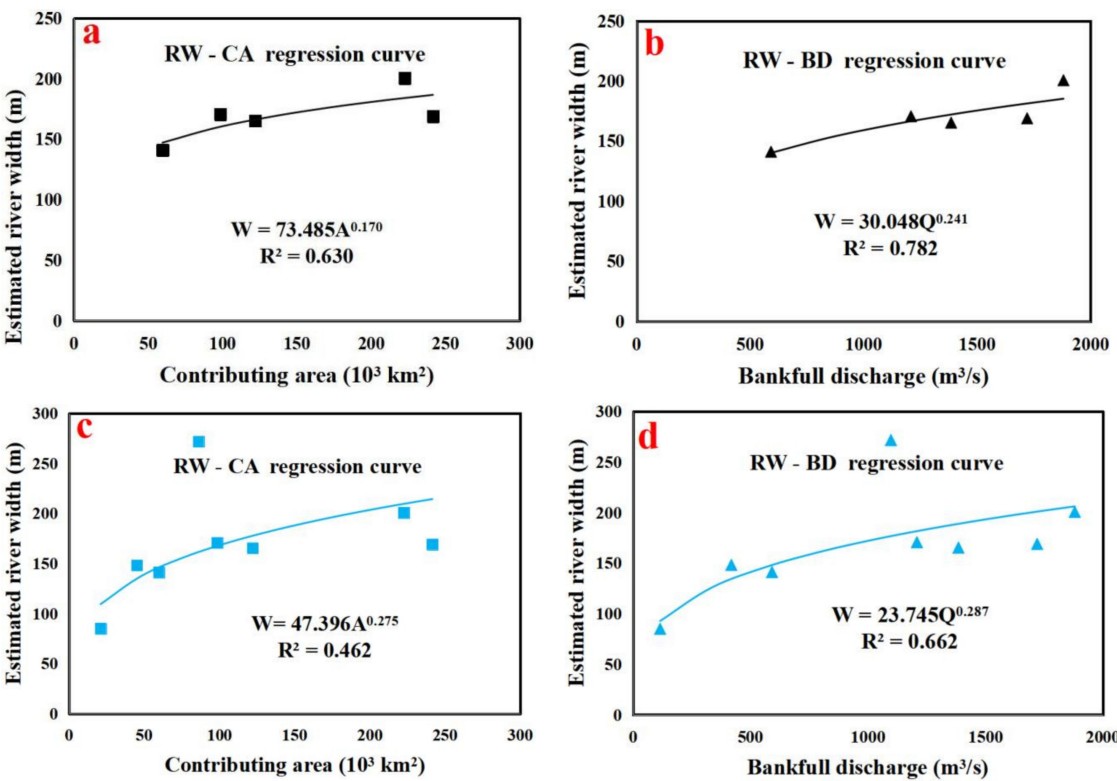

**Figure 10.** Regression relationships between the estimated river widths and the contributing areas and bankfull discharges. (**a**,**b**) The regressions of single-thread river reaches, excluding the hydrological stations affected by reservoirs; (**c**,**d**) the regressions of all river reaches, excluding the hydrological stations affected by reservoirs.

## 5. Discussions

In this study, the main factors that affected the accuracy and completeness of the river extraction are summarized as follows. First, the spatial resolution of the image directly determines the effective river width that can be measured. The effective width of the extracted rivers is 3 times greater than the image resolution. For small and narrow rivers, higher-resolution images must be used. The second factor is the quality of the satellite imagery. The cloud and shadow cover on the image directly affect the effective extraction of rivers. In the future, more effective algorithms must be developed for cloud and shadow removal. Third, the suspended sediment concentration of rivers and the water quality environment should be considered. The river's suspended sediment concentration is too high, and the water quality environment is complex, making the difference between the river's spectral and textural characteristics and the background small, and the extraction results are too noisy. Suspended sediment concentration and water quality are important factors affecting the extraction results.

In addition, river width distribution under bankfull discharge was impossible to obtain from the sparsely distributed hydrological stations, otherwise a remote sensing technique needs to be used. The study of the morphologies of alternatively distributed single-thread and multithread rivers can benefit from the extracted bankfull river widths based on the method developed in this study. Results indicate that the estimated river widths can be used as substitutes for the in situ-measured river widths when analyzing downstream hydraulic geometry. The good downstream hydraulic geometry relationship shows the connection of channel geometry of single-thread river reaches along the river course, though they are separated by multithread reaches. Both the geological and geomorphic background and inflow water and sediment contribute to the formation of river morphology. Results of this study indicate that inflow water and sediment contribute more to the morphology shaping of single-thread river reaches. By understanding the variation in bankfull river widths

along river reaches and across river networks, the effects of discharge and sediment on channel geometry can be predicted.

## 6. Conclusions

This work developed a method of extracting bankfull river widths on small rivers (width < 90 m) in mountainous areas based on remote sensing images and DEM, and preliminary explored the downstream hydraulic geometries of the main stream of the upper Yellow River. The main conclusions of this study are described as follows:

(1) The ML method exhibits good performance in the extraction of rivers in the upper Yellow River, and the extraction integrity can reach order 3 and above for the DEM drainage network. The mean overall accuracy of three subregions was above 0.87, and their mean kappa values were all above 0.75. The estimated $R^2$, RMSE, and MBE of the bankfull river width are 0.991, 7.455 m, and −0.232 m, respectively.

(2) Bankfull river widths of the mainstream were extracted with a step length of 5 km from the source to the exit. The average river widths of the single-thread sections showed a good linear relationship, with an $R^2$ value reaching 0.801. There are good power relationships between the river width and the bankfull discharge and contributing area, with $R^2$ values of 0.782 and 0.630, respectively.

(3) The effective connected river width was 30 m, which was 3 times the image resolution. The research results could enrich the river channel width database of the upper Yellow River and provide basic data for applications in hydrology, fluvial geomorphology, and stream ecology.

(4) The high spatial resolution of the bankfull river width dataset can be used to (1) compensate for the missing river width data between two traditional hydrological stations, and further analyze the channel geometries of alternatively distributed single-thread and multithread rivers; (2) analyze downstream hydraulic geometry and estimate bankfull discharge in river sections without hydrological data; (3) provide additional boundary conditions for distributed hydrological models to improve the simulation accuracy; and (4) quantify water carbon emissions [60].

The extraction of rivers below 30 m is relatively poor using the 10-m remote sensing images. In addition, due to the limitation of hydrological data and remote sensing images under bankfull discharge, only 22 hydrological stations were used in this study, and the research results had certain limitations. In the future, radar data will be combined with optical images with resolutions of 2 m or more to explore the automatic extraction of mountainous rivers with widths less than 30 m under complex terrains, to obtain the river width parameters of the whole river network. Simultaneously, the nature of the water body and the influence of substances in the water combined with the hydrological model will be considered to improve the accuracy of water body extraction in complex environments.

**Supplementary Materials:** The following are available online at https://www.mdpi.com/article/10.3390/rs13142650/s1, Table S1: Basic information of the cross sections located in the upper Yellow River Basin.

**Author Contributions:** Conceptualization, D.L. and B.W.; data curation, D.L., G.W. and C.Q.; funding acquisition, B.W.; investigation, G.W. and C.Q.; methodology, D.L.; software, D.L.; supervision, D.L. and B.W.; validation, G.W.; visualization, C.Q.; writing—review and editing, C.Q. and B.W. All authors have read and agreed to the published version of the manuscript.

**Funding:** This research was funded by the Natural Science Foundation of China, grant number 51639005 and 52009061, the National Key R&D Program of China, grant number 2017YFC0405202, and the Postdoctoral Innovation Talents Support Program of China, grant number BX20190177. The APC was funded by the Natural Science Foundation of China, grant number 51639005.

**Data Availability Statement:** Data provided by the Bureau of Hydrology at the Ministry of Water Resources of China were in the form of hard copy but not electronic copy, therefore, no link (URL or DOI) can be presented here. The other data and extraction results are available on request from the corresponding author.

**Acknowledgments:** The authors would like to thank Bowei Chen from the Institute of Remote Sensing and Digital Earth, Chinese Academy of Sciences for his guidance and help in compiling and modifying the program. And to acknowledge the Bureau of Hydrology at the Ministry of Water Resources of China for providing the in-situ measured hydrological data.

**Conflicts of Interest:** The authors declare no conflict of interest.

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
