# Peer review of "River Extraction under Bankfull Discharge Conditions Based on Sentinel-2 Imagery and DEM Data"

_remotesensing, doi:10.3390/rs13142650_

Round 1
Reviewer 1 Report
In the manuscript, the main focus is made on the methodological side of the question. The process of obtaining data using remote sensing is described in detail. However, the scientific and practical significance of the obtained results are not considered (although it is stated). How can the results be used in fluvial geomorphology or environmental management? Only as quantitative data source? There is no interpretation of some obtained results. For example, what explains the position of the curve (Fig. 9b) of the contributing area above the bankfull discharge on one segment and below the bankfull discharge on another segment? What can be said about changing the width and discharge along the Yellow river from the position of evolution and the forecast of the development of the riverbed?
It is necessary to correct the boundaries of the subregions (fig. 2), because in the present form, some subregions have borders of different colors.
Author Response
Many thanks for your advice. We provide a point-to-point response, please see the attached file.

Reviewer 2 Report
Review of Manuscript: "River Extraction under Bankfull Discharge Conditions Based on Sentinel-2 Imagery and DEM Data" by Dan Li and others
This manuscript uses ML-RF method to estimate bankfull river width based on Sentinel-2 Imagery and DEM data on the upper Yellow River. It seems to me to be a novel approach to get key geomorphological data on rivers where width is between 30-90 m. The produced data are very important for understanding the fluvial geomorphology on the upper Yellow River. However, the results are accessed based on only 7-8 measured data points, on a river network with >2000 km of mainstem. The other concern is sediment concentrations in the studied are not high at all, particularly for the Yellow River, though the variation of the sediment concentration is large.
English can be improved. Many wordy and repetitive sentences...
Editorial suggestions are attached in the PDF comments.

Author Response

(The authors gave the same response as above.)

Reviewer 3 Report
Dear Authors,
Thank you so much for the updated manuscript submission to Journal of Remote Sensing. After careful review, I believe that this paper presents good set of work, and the comprehensive quality has be significantly improved. To ensure high quality of techinical paper coming into RS journal acceptance, I summarized some possible issues (not limited to), which are listed as follows:
a) Abstract session contains about 400 words, which seems too long. Consider shorten this session into 300~320 words in the final version.
b) Emphasize your keynote contributions right after Figure 1 in the Introduction Section. The last paragraph is preferable with organization on the remainder of this paper. (If you have no section on Related Work, this set of opinion is for your reference. Thanks.)
c) In the current version, Table 2 takes up the whole Page 7, while Table 3 cross over Pages 10-11. I suggest the authors consider compact format to shape the updated Tables (i.e., restrict Table 3 in the same page, in double-spaced version), which may have better appearance and the same way to propagate your tabulated results.
d) Rearrange Figure 5 in 2 x 3 subdiagrams with higher resolutions in legends; be sure that all the rest figures have no distortion or occlusions.
e) In the figures which has any quantitative results of R^2, please consider rounding all the decimals to 3 valid digits (i.e., like R^2 = 0.991 in Figure 8 (the footnote should be shifted to the bottom of Page 15), in Figure 10, the four R^2 should be presented as 0.630, 0.782, 0.463, and 0.662, respectively).
f) Conclusion section: consider making the four sets of keynote conclusion statements more concise, and add some more specific details on limitations of study and suggested future work (in the last paragraph).
g) References: check with Remote Sensing journal to see whether the abbreviated terms on citations are preferrable, then be sure the cited refereces are in professional shape; adding some more related latest work got published in Years 2019-2021.
f) Overlap measure with Ref. [20]: the authors presented similar work on river extraction using technical machine learning based approach for Sentinel Imagery and DEM data. I advise the authors to measure the overlap ratio between your manuscript and [20], which should be less than 30%.
g) There are minor language and grammatical issues persisting in the current version, consider inviting a native speak to proofreading the whole set of manuscript then re-submit the polished article to Remote Sensing.
I recommend this manuscript with a minor revison before applying review comments, then consider coming into acceptance. Once again, thank you very much, and good luck with your further edits!
Best wishes,
Yours sincerely,
Author Response

(The authors gave the same response as above.)
